# Telemedicine Improves HCV Elimination among Italian People Who Use Drugs: An Innovative Therapeutic Model to Increase the Adherence to Treatment into Addiction Care Centers Evaluated before and during the COVID-19 Pandemic

**DOI:** 10.3390/biology11060800

**Published:** 2022-05-24

**Authors:** Valerio Rosato, Riccardo Nevola, Vincenza Conturso, Pasquale Perillo, Davide Mastrocinque, Annalisa Pappalardo, Teresa Le Pera, Ferdinando Del Vecchio, Ernesto Claar

**Affiliations:** 1Liver Unit, Ospedale Evangelico Betania, 80147 Naples, Italy; riccardo.nevola@unicampania.it (R.N.); pasqualeperillo@hotmail.it (P.P.); davidemastrocinque4@gmail.com (D.M.); annalisa.pappalardo88@gmail.com (A.P.); ernestoclaar@gmail.com (E.C.); 2DS32 (Distretto Sanitario N. 32), Ser.D. Unit, ASL Napoli 1 Centro, 80147 Naples, Italy; enzaconturso@gmail.com (V.C.); teresa.lepera@fastwebnet.it (T.L.P.); ferdelv@libero.it (F.D.V.)

**Keywords:** hepatitis C, person who uses drugs (PWUD), differentiated model of care, COVID-19

## Abstract

**Simple Summary:**

People who use drugs represent a category of patients to be prioritized for antiviral treatment for the purpose of hepatitis C elimination, due to their younger age and the major risk of viral transmission, acting as a virus reservoir. The treatment challenges for hepatitis C in this population are related to an ineffective linkage to care, poor adherence to treatment, and follow-up and the risk of re-infection. The COVID-19 pandemic has further exacerbated these conditions, increasing the concerns among clinicians regarding the effectiveness of their treatment. In our study, we describe a novel “patient-tailored” model-of-care for people who use drugs. The antiviral therapy was adapted to the needs of the patient and monitored remotely by a hepatological specialist, in order to decentralize the point of care within the addiction center. The study was conducted before and during the COVID-19 pandemic, clearly demonstrating the model’s high effectiveness in the linkage to care, adherence, and response to antiviral therapy.

**Abstract:**

People who use drugs (PWUDs) are generally considered “hard-to-treat” patients, due to adherence to HCV antiviral therapy or re-infection concerns. Linkage-to-care still remains a significant gap for HCV elimination, worsened by the COVID-19 pandemic. To reduce time-to-treat and improve treatment adherence, we have developed a patient-tailored model-of-care, decentralized within the addiction center and supervised remotely by hepatologists. From January 2017 to December 2020, patients were enrolled in one addiction care center in Southern Italy, where a complete hepatologic assessment, including blood chemistry, ultrasound, and transient elastography examination, was provided. DAAs treatment has been adapted on clinical features, also performing a daily administration during an outpatient visit, and monitored remotely by specialists via telemedicine interactions. Adherence was evaluated on the accomplishment of therapy or on the percentage of attended visits. From a total of 690 PWUDs, 135 had an active HCV infection and were enrolled in the study. All patients started the treatment within 3 weeks after HCV diagnosis. Six drop-outs were recorded, obtaining a sustained virological response at week 12 (SVR12) in 98.5% of PWUDs. There were only two cases of treatment failure, one of which is re-infection. No differences were found between the SVR12 rates before and during the COVID-19 pandemic. We obtained a high SVR12 rate, providing a comprehensive assessment within the addiction care center, tailoring the drug administration with a hepatologic remote stewardship. Our therapeutic model should improve the time-to-treat and treatment adherence in PWUDs.

## 1. Introduction

The recent availability of extremely safe and pangenotypic direct-antiviral agents (DAAs) has allowed the potential for a foreseeable hepatitis C virus (HCV) elimination. The 69th world health assembly, planned for 2030, is a goal of the World Health Organization (WHO) [1]. Nevertheless, it has been reported that, among high-income countries, only nine are on track to reach this target, while about 30 countries are not projected to eliminate HCV before 2050, mostly due to a lack of HCV screening, linkage-to-care, and treatment strategy [2]. It has been proposed as the most pragmatic approach to reach global HCV elimination, a micro-elimination strategy, consisting of breaking down the national elimination goal into smaller goals for well-defined population segments, achieving more quick and efficient screening and treatment interventions [3]. Among the 56.8 million people with hepatitis C infections globally [4], about 39.2% are people who use drugs (PWUDs), and about 8.5% of new HCV infections occur in PWUDs [5].

Usually, PWUDs are younger compared to the other common categories of HCV patients (i.e., baby boomers), with a major risk of HCV transmission, acting as virus reservoirs [6]. In Italy, in 2018, an incidence of HCV of 5.83/100 person-years on a cohort of 284 uninfected PWUDs followed prospectively for 12 months was estimated [7]. In this type of population, the achievement of a high rate of sustained virological response (SVR) is compromised by the poor adherence, reduced tolerability, and risk of HCV reinfection. During the era of the interferon-based therapeutic regimen, these barriers may explain the reluctance of clinicians in the prescription of HCV treatment [8]. In 2012, in 25 addiction care units, on a population of 543 PWUDs, a prevalence of HCV-Ab of 63.9% was estimated, but only 19.3% of these patients received antiviral treatment [8]. The DAAs treatments are effective in PWUDs and in people receiving opioid substitution therapy (OST), even in “real-world” settings, reaching a rate of SVR that exceeds 90% [9]. It has been showed that the management of HCV treatment inside the addiction care units, in Italy named SerDs, may improve the performance of DAAs, providing pharmacological and psychological treatment, with the aim of allowing a social rehabilitation of PWUDs [10].

A major obstacle to HCV cure in this population is lack of an effective linkage between peripheral SerDs and prescribing centers. Although these structures may be very far apart, the dislocation of assistance in various centers, rather than the real geographical distance, represents the gap in the HCV elimination pathway [11].

Furthermore, the COVID-19 pandemic disrupted healthcare service delivery, having a deep impact on chronic liver disease management due to the shift to telemedicine and reduction or elimination of in-person clinical encounters. PWUDs have been disproportionately affected by this change in healthcare system, not only due to a 10-fold increased risk of acquiring COVID-19 [12], but also to a low access to necessary treatment service for OST or other chronic infectious disease [13]. Comparing with the 2018 and 2019, in the early COVID-19 pandemic, screening for and treatment of HCV markedly decreased, falling by 62 and 43% respectively, thus potentially leaving PWUDs with undiagnosed and untreated infection [14]. The American Association for the Study of the Liver Diseases (AASLD) and the European Association for the Study of Liver (EASL) advocated simplifying HCV treatment algorithms, limiting staging and treatment surveillance to achieve the WHO elimination goal [15].

From this perspective, we describe an innovative HCV treatment mode for PWUDs that improves the linkage-to-care and exemplifies the patient monitoring during antiviral therapy.

In a single SerD of Southern Italy, since 2017 we applied this “patient-centered approach” to the HCV treatment in PWUDs, that showed, even in the pre-COVID era showed, excellent efficacy in PWUDs management, and especially during the COVID-19 pandemic, demonstrated further potential in micro-elimination strategy.

## 2. Materials and Methods

### 2.1. Description of the Innovative Model of Treatment

This is an observational prospective monocentric study conducted on PWUDs. Our model has been applied in a single SerD of Southern Italy, located on the outskirts of Naples in one of the poorest neighborhoods with a high illiteracy rate. The distance between the SerD and the nearest prescribing center is 3 km, but the ineffectiveness of public transport, the complexity and the duration of baseline assessments prior to treatment represented a major impediment at the onset of HCV treatment, which could be delayed for several months or, sometimes, canceled. Usually, the patients screened positive for HCV often required 1–2 specialist visits to the clinician in-person to evaluate them and initiate treatment. Therefore, our model consisted primarily of providing a comprehensive hepatological assessment within the SerD for PWUDs, assessing the eligibility for HCV therapy and initiating the antiviral treatment after a single in-person specialist visit. In this way, the patients, following diagnosis of HCV infection by SerD physicians and any additional diagnostic test for chronic disease agreed in telemedicine with hepatologists, were evaluated by a specialist within 3 weeks, starting the antiviral treatment immediately after the in-person visit. In response to the different needs and behavioral difficulties of this particular category of patients, the subsequent DAAs treatment was adapted according to clinical or social features of patients under the supervision of SerD physicians. So, in patients deserving of enhanced support, weekly or more frequent phone calls were provided and/or a flexible directly dispensing of the DAAs medication, either weekly or daily, were performed by the SerD’s staff. The administration of antiviral therapy often occurred at the same time as that of OST within the SerD. After the start of treatment, the patients were followed by a hepatologist exclusively through telemedicine interaction (mostly phone calls) and, after the obtaining of SVR, the subsequent in-person visits for hepatological follow-up were reserved only to selected patients. Patients deserving of specialistic follow-up were selected on clinical features, including advanced liver fibrosis, focal liver lesions, NAFLD or other comorbidities. The graphic description of our “patient-centered approach” to HCV treatment in PWUDs was reported in Figure 1.

### 2.2. Study Setting

The study started in January 2017 and was conducted until December 2020. We considered as PWUD all individuals who have taken injection or non-injections drugs (i.e., by inhalator route) in the last 12 months, including people receiving OST. All clients who accessed our addiction care center in southern Italy were screened for HCV and those who screened positive for HCV antibody (HCV Ab) underwent further evaluation with testing for HCV-RNA and HCV genotype, after a median wait of 1 week. All consecutive PWUDs with active HCV infection were enrolled in the study. Moreover, all patients were screened for HIV or chronic HBV infection and women underwent beta HCG testing for pregnancy. All the blood samples for the laboratory tests were collected within the SerD and, subsequently, sent to the analysis laboratory by dedicated personnel. During the first live specialist evaluation performed within the SerD after a median wait time of 2 weeks, the hepatological assessment was completed with ultrasound examination and evaluation of hepatic fibrosis. We used liver stiffness (FibroScan transient elastography) to assess the stage of liver fibrosis and the chosen cutoffs of clinical significance were 7.1–12.9 kPa for moderate or advanced liver fibrosis and ≥13 kPa for cirrhosis. The presence of cirrhosis was also defined on compatible clinical (ascites, esophageal varices), laboratory (platelet < 100.000/mm^3^, albumin < 3.5 or INR > 1.7) or ultrasounds (coarse pattern, irregular liver surface and evidence of portal hypertension, such as splenomegaly) features. Demographic characteristics and clinical parameters at baseline, substance abuse, current working condition and the OST were also recorded. All stages of liver disease suitable for the antiviral treatment and HBV coinfection were included in the study. Treatment naïve patients and experienced ones were enrolled indifferently. Instead, patients with diagnosis of hepatocellular carcinoma, pregnant women or HIV coinfection were excluded. The adherence to treatment was evaluated by SerD physicians based on the therapy accomplishment or on the percentage of attended visits. After completion of therapy, the sustained virological response was evaluated at week 12 (SVR 12) and, when required, re-infection was assessed by genotyping.

### 2.3. Antiviral Therapy

Patients underwent DAAs treatment following international guidelines [15,16] and the eligibility Drug Agency Committee with the last update in March 2017 [17]. Patients were only treated with third-generation DAAs: glecaprevir/pibrentasvir (GLE/PIB) for 8 weeks or sofosbuvir/velpatasvir (SOF/VEL) for 12 weeks. Ribavirin was added in cirrhotic genotype 3 patients. The choice of antiviral therapy was based on severity of liver disease, comorbidities, potential pharmacokinetic drug interactions and patient preferences for pill burden and treatment duration.

### 2.4. Study Outcomes

The primary outcome of the study was to assess the effectiveness of our model of treatment in PWUDs by detection of SVR 12 and adherence to DAA treatment. Also, the characteristics of poor adherence patients, treatment completion, serious adverse events and HCV reinfections were evaluated. Additional outcome was to evaluate the impact of the COVID-19 pandemic on the efficacy of our model of care.

### 2.5. Statistics

Continuous variables are expressed as median values and interquartile range (IQR), and categorical data represent the frequencies and percentages. The data normality was checked by Shapiro-Wilk test and the Student *t*-test or the Mann–Whitney test were performed appropriately to compare differences in the values of continuous variables. Fisher’s exact test and the chi-square test with the Yates correction were used to evaluate the significance of associations among categorical variables. Statistical significance was defined as *p* < 0.05 (two-tailed test, 95% confidence interval). Statistical analyses were performed using the SPSS software (SPSS version 20, SPSS Inc., Chicago, IL, USA).

### 2.6. Ethics

The study was conducted in accordance with the guidelines of the Declaration of Helsinki and the principles of good clinical practice. The patient’s data were collected and conserved according to the general data protection regulation (EU) 2016/679. Informed consent was obtained from all subjects involved in the study. The study was approved by our Institutional Review Board.

## 3. Results

### 3.1. Study Sample

During the study period, 690 PWUDs were tested for HCV Ab, including patients already followed at the SerD and new members. 28 PWUDs, already followed at the SerD, had previously been treated and were therefore not tested for HCV Ab. Of all patients tested, 309 HCV Ab positive PWUDs were identified (44.8%) and 135 had a detectable HCV RNA (43.7%).

The characteristics of the population with active HCV infection were reported in Table 1. The patients are mainly male, with a median age of 50 years. The genotype 1a and genotype 3 were the most represented. A total of 22 patients (16.3%) were cirrhotic, while 27 (20%) had advanced fibrosis. 96.2% of patients were treatment naïve, the rest (5 PWUDs) had previously been treated with Peg-interferon + ribavirin. No patients had previously been treated with DAAs. 58 PWUDs expressed HBcAb positivity, but only 2 patients had a concomitant active HBV infection, both receiving NUCs. No HDV coinfection was found. 3 HIV co-infections have been identified, but HIV patients are treated in other referral centers in our area, so they were excluded from the study. Most patients used intravenous drugs (114, 84.5%), but a small portion had exclusively a history of intranasal drug use or smoking. This data proves how all PWUDs are at risk of HCV. Anyway, 67.4% of patients were active drug users and 74 patients (54.8%) were currently employed. Moreover, 85 PWUDs (62.9%) were routinely followed within the SerD and were undergoing contemporaneity treatment with OST (66 with methadone and 19 with buprenorphine). Approximately 22.9% of patients were concomitantly using benzodiazepines.

### 3.2. PWUDs Linked to Care and Response to Antiviral Treatment

All patients with active HCV infection accepted to start treatment. The main DAA regimen used was GLE/PIB (53.4%), while 63 PWUDs were treated with SOF/VEL and in 7 of them ribavirin was added. Most patients with genotype 3 infection and with cirrhosis were treated with SOF/VEL. GLE/PIB regimen was mostly used in PWUDs without liver fibrosis or with mild fibrosis and in patients with genotype 1b infection. Furthermore, SOF/VEL treatment was mainly administered in patients concomitantly using benzodiazepines or other psychotropic drugs. No serious adverse events occurred during the treatment.

Six patients (4.4%) dropped out of the treatment during the first month of antiviral therapy, but, despite interrupting the therapy, only one of them did not achieve a sustained virological response. A single re-infection has been observed at week 12 post-treatment in a PWUD, active drug user, who successfully completed treatment with GLE/PIB regimen. Overall, although 129 patients completed the antiviral treatment, an SVR 12 was obtained in 133 patients (98.5%). The HCV care cascade is shown in Figure 2. The rate of SVR 12 was not different between SOF/VEL and GLE/PIB regimens, 98.4 and 98.6% respectively.

### 3.3. Adherence to HCV Treatment

Among our patients, 9 of them did not show adequate adherence to treatment, failing more than 90% of scheduled visits during antiviral therapy (>1 visit). The characteristics of these patients were reported in Table 2. Eight patients were active heroin users and 3 were on OST with methadone. 6 PWUDs dropped out of the treatment during the first month of therapy. Five were treated with SOF/VEL and four with GLE/PIB. As already mentioned, of these patients only 1, treated with SOF/VEL, did not achieve the SVR12. The other three PWUDs with poor adherence completed the antiviral therapy, even if they did not respect the scheduled visits, delaying withdrawal of the drug packs and then temporarily interrupting the antiviral therapy for a few days. Nevertheless, all these patients achieved virological response at SVR 12. Consequently, 96.9% of our patients completed the antiviral therapy, even if 6.6% of patients could be defined as non-adherent to treatment, but, despite all, these patients showed a rate of SVR 12 of 88.8%.

### 3.4. Antiviral Therapy during COVID-19

Among our population, 25 PWUDs underwent antiviral treatment during the COVID-19 pandemic. The characteristics of these patients are shown in Table 3. Compared to the 110 PWUDs treated before COVID-19, the patients treated during the pandemic had a lower prevalence of males. Nevertheless, no other statistically significant differences were found. Furthermore, all patients treated during COVID-19 achieved SVR12 and no one dropped out of the treatment. Poor adherence to treatment was recorded in only 2 cases, but both patients successfully completed the antiviral therapy.

## 4. Discussion

In our study, we describe a “patient-tailored” model of care for HCV in the cohort of PWUDs implemented by telemedicine-based hepatological stewardship. In our population, we obtained an excellent SVR 12 rate of 98.5% with the 100% of linkage to care.

PWUDs represent a category of patients to be prioritized for antiviral treatment for the purpose of HCV micro-elimination, but the treatment rates have been historically disappointing due to their poor compliance to treatment [18]. But these considerations were mostly carried out when interferon-based treatment was standard of care. In fact, the universal access to DAAs regimen allowed to achieve high SVR 12 rates as reported by several studies on PWUDs performed in real world settings [10,11,19,20]. Nevertheless, in most countries of world, including Italy, access to DAAs remains extremely limited among PWUDs. In 2019 a survey that involved the physicians of 30.3% of SerDs in Italy, showed that <50% of PWUDs received treatment, due to nurse understaffing and technical, economical and logistic issues [21]. So, simplifying the model of care is a key aspect to achieve the WHO target of HCV elimination in 2030. First of all, the integration of all services into one point of care seems to be fundamental to effective treatment of PWUDs [22]. Given the chaotic lifestyle of these individuals, the HCV point-of-care treatment should be decentralized to areas where PWUDs are already have access to other services, such as OST or syringe distribution [23]. In a recent Spanish study, the implementation of HCV point-of-care “test and treat” strategy has shown to increase the access to treatment and SVR rates, reducing the reinfection and loss to follow-up rates, compared to the standard-of-care [24]. Furthermore, the COVID-19 pandemic has required adaptations in providing clinical service, so in our model we decentralized all phases of HCV treatment into the SerDs, reducing the patient’s access to multiple healthcare facilities and, consequently, minimizing the in-person visits. At the time of HCV diagnosis, a multidisciplinary assessment of patients was ensured by interactions in telemedicine, in order to integrate other investigations, such as laboratory tests, and to guarantee the beginning of therapy immediately after a single in-person specialist visit. In our study, all patients with active HCV infection started the antiviral therapy, so our treatment rate appears slightly higher than those reported in other previous studies in similar settings, ranging from 67 to 99.1% [11,20,24].

The SOF/VEL regimen was chosen based on the presence of psychotropic co-medications or the SVR rate expected in patients with genotype 3 or advanced liver disease, while the other patients were treated with an 8-week GLE/PIB regimen. Ribavirin was used in combination with SOF/VEL in only a few patients treated before 2018. No side effects or drug interactions were reported, despite 22.9% of patients were taking psychotropic medications. This finding confirms the high tolerability and the safety of SOF/VEL regimen, as already demonstrated in MINMON study, in which it was observed that a minimal monitoring approach (only clinical baseline assessment and SVR laboratory testing) may be safe and effective as the standard monitoring approach [25].

Treatment discontinuation and loss to follow-up are the main concerns about the efficacy of HCV treatment in PWUDs. Thanks to the assessment of SerD’s physicians, PWUDs are not only supervised for the OST but they also undergo clinical and psychological evaluations. It has been demonstrated by Rinaldi et al. that the SerD management allows to reach a higher SVR rate in PWUDs compared to patients not followed by addiction centers, reducing the dropouts and increasing the adherence to treatment [10]. So, providing a treatment tailored on clinical e social features of patients seems to be the most effective strategy to treat this vulnerable population. In our study, once antiviral therapy began, the frequency of visits and monitoring of patients was managed by physicians and nurse staff of SerD, who were more aware of the patient’s needs. In some cases, DAAs treatment was administered daily in conjunction with the delivery of OST. 96.9% of our patients completed the treatment with only 6 drop-out, showing a strong adherence to therapy given that approximately 67.4% of participants were active drug users. Our results are well in keeping with those reported in the recent study of Mangia et al., in which 229 PWUDs (74.8% active drug users) have been treated with a model of care based on personalized interventions. 95.6% of patients completed the treatment and drop-out was recorded in only 9 cases [11]. Furthermore, delivering the DAA therapy within the SerD certainly increased linkage to care and adherence to treatment in the frail patient category, as shown in the Australian study by Read et al. [26]. In this study conducted in Kirketon Road Centre in Sydney, among 72 PWUDs treated for HCV, 25 required an enhanced adherence-support package, including weekly or daily administration of DAA at the addiction center, obtaining an SVR 12 of 92% [26].

In our populations, only two patients did not achieve the SVR 12, despite six patients dropping out of the treatment, and three patients showing poor adherence to the scheduled visits. Eight of these people were active drug users. Even considering the drop-out at first month or the temporary interruption of the treatment, none of these patients were lost to follow-up and only 1 of these PWUDs did not achieve the SVR12. These findings are in line with those of the SIMPLIFY study, underlining that most non-responses reported in phase 3 clinical trials on PWUDs were related to loss to follow-up and not to virological failure or relapse [27]. In fact, in SIMPLIFY study the adherence did not significantly affect the SVR that was 94%, despite a high lack of adherence to treatment, especially in recent injection drug users (66%) [27].

Twenty-five patients have been treated during the COVID-19 pandemic and no treatment failures have been reported. Our model of care, based on telemedicine communications, did not require patients to have access to healthcare facilities other than the SerD. All the treatment was performed under a hepatological stewardship in telemedicine, sending laboratory examinations periodically by emails and making phone or video calls. Despite 49 patients having advanced liver disease, during the treatment no PWUDs required to access to hospital facilities or other in-person visits, beyond the first at the start of antiviral treatment. In a recent study of Sivakumar et al., conducted in New Haven, USA, during the early months of COVID-19 pandemic, 31 active injecting PWUDs have been treated with a simplified treatment algorithm consisting of minimal in-person visits during the service syringe program followed by telehealth communication, primarily telephone calls, between patients and practitioners or clinicians [28]. All patients in this study completed the treatment and 29 patients achieved the SVR12 (93.5%), confirming that, alongside the type of patients who need an enhanced in-person monitoring, many PWUDs are able to self-manage their care adequately. Recently, in a Ukrainian report, it has been shown that during COVID-19 pandemic, despite an increase in delivering of take-home dosing to over 82.2% of patients, the mortality and overdose cases in PWUDs did not increase [29]. These findings may support the use of telemedicine interaction even after COVID-19 case rates decrease and patients can resume in-person visits, reducing the need for patient travel and implementing the interactions between specialists for multidisciplinary management of PWUDs. Furthermore, the implementation of telemedicine in the model of care could be crucial for PWUDs who may not be able to maintain steady access to regular outpatient visits due to socioeconomic factors (i.e., housing instability) or to clinical and social features (i.e., mental disorders, stigmatization by healthcare system). Our model of care showed high efficacy both before and during COVID-19 pandemic, suggesting its future possibility to improve the HCV micro-elimination in PWUDs. In addition, HCV treatment should also be considered as a preventive measure for PWUDs, given that actively injecting drug users are the main contributors to continued transmission [30].

Despite these important findings, this study has some limitations. Firstly, the study was conducted in a single location which is a very long-standing SerD with a staff widely specialized in the management of complex and vulnerable individuals as PWUDs. Unfortunately, unlike other settings, dealing with PWUDs often requires skills and experience that can only be acquired through extensive experience in this field and through an operator’s marked sensitivity to the social problem of the patients. Therefore, further extensive studies are needed to establish the breadth of contacts and the experience of nurses and physicians of SerD needed to maintain an optimal care for PWUDs. In this study, the telemedicine interactions were performed mainly via phone calls, which were sufficient to contact the patients, while the communication between SerD physicians and hepatologist specialists also took place by e-mail. In other settings, the availability of this type of communication could be lacking (i.e., rural settings), therefore implementing the access to telemedicine may represent a further challenge.

This study was conducted from January 2017 to December 2020, a time frame in which not only the treatment of hepatitis C changed slightly, but our social habits changed too due to the COVID-19 pandemic. However, despite these changes, the effectiveness of our care model has not changed. This finding could represent a strength of our study, but further evidences are needed to confirm the reproducibility of our model of care.

## 5. Conclusions

This study provides evidence on the effectiveness of our innovative model of care based on telemedicine for HCV treatment in PWUDs in a real-world setting. Adapting the HCV micro-elimination strategies to the needs of patients may be more effective than the simply application of the international guidelines. The collaborations between the different specialists, as hepatologists, and the SerD physicians may be effectively ensured by telemedicine communications, decentralizing the point of care of PWUDs. Furthermore, the implementation of telemedicine in the management of these hard-to-treat patients may allow for HCV eradication despite exceptional and unexpected events, such as the COVID-19 pandemic.

## Figures and Tables

**Figure 1 biology-11-00800-f001:**
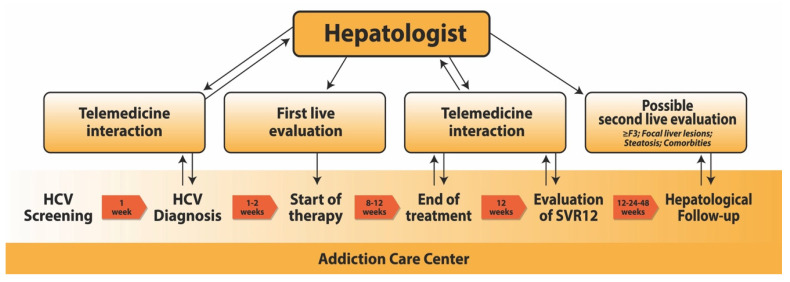
Graphic description of our “patient-centered approach” to HCV treatment in PWUDs.

**Figure 2 biology-11-00800-f002:**
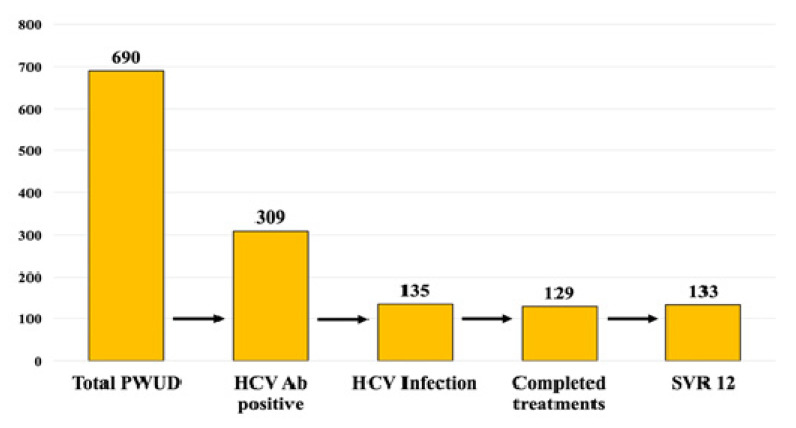
HCV care cascade. 309 PWUDs, 44.8% of total patients tested, were HCV Ab positive and of this, 135 (43.7%) had a detectable HCV RNA. All patients with an active HCV infection started the antiviral treatment, but 6 patients dropped out in the first month. Despite the interruption of the treatment, none of these patients were lost to follow-up and only 1 did not achieve the virological response at week 12 post-treatment. Another patient experienced a reinfection, so the SVR 12 rate was of 98.5%.

**Table 1 biology-11-00800-t001:** Clinical features of PWUDs treated matched by DAAs regimen.

Variables	Total *n.* 135	SOF/VEL *n*. 63 (46.6)	GLE/PIB *n*. 72 (53.4)	*p* Value
Age, years	50 (46–55)	51 (46–56)	50 (46–54)	0.290
Male	129 (95.6)	62 (98.4)	67 (93)	0.134
Currently employed	74 (54.8)	35 (55.5)	39 (54.1)	0.873
Ways of drug administration				
intravenous	114 (84.5)	61(96.8)	64 (88.8)	0.137
intranasal	8 (5.9)	1 (1.6)	7 (9.7)	0.068
smoke	13 (9.6)	4 (6.3)	9 (12.5)	0.148
Active drug users	91 (67.4)	52 (82.5)	39 (54.1)	0.205
Currently in OST	85 (62.9)	46 (73)	39 (54.1)	0.022
Methadone	66 (48.9)	35 (55.5)	31 (43)	0.149
Buprenorphine	19 (14)	11 (17.4)	8 (11.1)	0.293
Benzodiazepine or other psychotropic drugs	31 (22.9)	20 (31.7)	11 (15.2)	0.023
HCV Genotype				
1a	60 (44.4)	25 (39.6)	35 (48.6)	0.111
1b	8 (5.9)	1 (1.6)	7 (9.7)	0.047
2	3 (2.2)	0	3 (4.1)	0.096
3	57 (42.3)	35 (55.5)	22 (30.5)	0.017
4	7 (5.2)	2 (3.2)	5 (6.9)	0.178
HBcAb	58 (42.9)	25 (39.7)	33 (45.8)	0.219
HBsAg	2 (1.4)	1 (1.6)	1 (1.3)	0.925
Alanine transaminase, IU/mL	53 (31–90)	47 (31–94)	54 (31–84)	0.987
Aspartate transaminase, IU/mL	41 (27–69)	45 (30–77)	38 (26–54)	0.248
Gamma glutamyl transferase IU/mL	50 (28–89)	55 (24–85)	48 (30–89)	0.702
Stage of liver disease *				
No or mild Fibrosis (F 0–1)	86 (63.7)	29 (46)	57 (79.1)	<0.001
Moderate or advanced fibrosis (F 2–3)	27 (20)	15 (23.8)	12 (16.6)	0.380
Cirrhosis (F 4)	22 (16.3)	19 (30.1)	3 (4.1)	<0.001
Treatment experienced (Peg-IFN + Ribavirin)	5 (3.7)	2 (3.2)	3 (4.1)	0.534
Type 2 Diabetes mellitus	9 (6.6)	5 (7.9)	4 (5.5)	0.317
Ribavirin	7 (5.1)	7 (11.1)	0	0.051
Adherence to treatment < 90%	9 (6.6)	5 (7.9)	4 (5.5)	0.384
Drop-out	6 (4.4)	4 (6.3)	2 (2.7)	0.868
Reinfection	1 (0.7)	0	1 (1.3)	0.453
SVR 12	133 (98.5)	62 (98.4)	71 (98.6)	0.925

Data are expressed as median (IQR) for continuous variables and *n* (percentage) for categorical variables. OST: opioid substitution therapy; SVR 12: sustained virological response at week 12. * evaluated with transient elastography: F 0–1 (<7.1 kPa), F 2–3 (7.1–13 kPa), F 4 (>13.1 kPa).

**Table 2 biology-11-00800-t002:** Characteristics of PWUDs with poor adherence to treatment (<90%).

N. of patients	9
Age, years	48 (43–55)
Male	9 (100)
Currently employed	2 (22.2)
Active drug users	8 (88.8)
Substance of abuse	
Heroin	9 (100)
Ways of drug administration	
intravenous	7 (77.7)
intranasal	1 (11.1)
smoke	1 (11.1)
Currently in OST	3 (33.3)
Methadone	3 (33.3)
Benzodiazepine	4 (44.4)
HCV Genotype	
1a	3 (33.3)
3	6 (66.6)
Stage of liver disease *	
No or mild Fibrosis (F 0–1)	8 (88.8)
Moderate or advanced fibrosis (F 2–3)	0 (0)
Cirrhosis (F 4)	1 (11.1)
Treatment	
Sofosbuvir + Velpatasvir 12 weeks	5 (55.5)
Glecaprevir + Pibrentasvir 8 weeks	4 (44.4)
SVR 12	8 (88.8)

Data are expressed median (IQR) for continuous variables and *n* (percentage) for categorical variables. OST: opioid substitution therapy; SVR 12: Sustained Virological Response at week 12. * evaluated with transient elastography: F 0–1 (<7.1 kPa), F 2–3 (7.1–13 kPa), F 4 (>13.1 kPa).

**Table 3 biology-11-00800-t003:** Clinical features of PWUDs matched for period of treatment.

Variables	Before COVID-19	During COVID-19	*p* Value
N. of patients	110	25	
Age	50 (46–55)	51 (43–56)	0.931
Male	107 (97.2)	22 (88)	0.042
Currently employed	61 (55.4)	13 (52)	0.759
Ways of drug administration			
intravenous	92 (83.6)	22 (88)	0.946
intranasal	7 (6.3)	1 (4)	0.376
smoke	11 (10)	2 (8)	0.258
Active drug users	75 (68.2)	16 (64)	0.690
Currently in OST			
Methadone	53 (48.2)	13 (52)	0.733
Buprenorphine	16 (14.5)	3 (12)	0.743
Benzodiazepine or other psychotropic drugs	25 (22.7)	6 (24)	0.131
HCV Genotype			
1a	52 (47.2)	8 (32)	0.171
1b	8 (7.2)	0	0.185
2	2 (1.8)	1 (4)	0.296
3	46 (41.8)	11 (44)	0.536
4	7 (6.3)	0	0.238
Stage of liver disease *			
No or mild Fibrosis (F 0–1)	71 (64.5)	15 (60)	0.376
Moderate or advanced fibrosis (F 2–3)	22 (20)	5 (20)	0.486
Cirrhosis (F 4)	17 (15.4)	5 (20)	0.284
Diabetes mellitus type 2	8 (7.2)	1 (4)	0.476
Treatment experienced with Peg-IFN + RIBA	5 (4.5)	0	0.291
Adherence to treatment < 90%	7 (6.3)	2 (8)	0.901
Drop-out	6 (5.4)	0	0.235
Treatment			
Sofosbuvir + Velpatasvir 12 weeks	48 (43.6)	15 (60)	0.141
Glecaprevir + Pibrentasvir 8 weeks	62 (56.3)	10 (40)	0.178
SVR 12	108 (98.2)	25 (100)	0.501

Data are expressed as median (IQR) for continuous variables and *n* (percentage) for categorical variables. OST: opioid substitution therapy; SVR 12: Sustained Virological Response at week 12. * evaluated with transient elastography: F 0–1 (<7.1 kPa), F 2–3 (7.1–13 kPa), F 4 (>13.1 kPa).

## Data Availability

Data are available at our Institute repository.

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
