# Peer review of "Telemedicine Improves HCV Elimination among Italian People Who Use Drugs: An Innovative Therapeutic Model to Increase the Adherence to Treatment into Addiction Care Centers Evaluated before and during the COVID-19 Pandemic"

_biology, 2022, doi:10.3390/biology11060800_

Round 1
Reviewer 1 Report
In their study, Rosato and co-workers analyze the impact of a telehealth approach to cure HCV among Italian people who use drugs.
The research context is very topical, especially given the WHO’s goal of eliminating HCV infection by 2030. Indeed, due to their younger age and the significant risk of viral transmission, people who use drugs act as virus reservoirs and are a problematic category of patients to be tested, linked to care, treated, receive appropriate follow-up, and have equitable access to care.
The study design and statistical analysis are appropriate. The authors adequately describe the methods and present the results that support their conclusion.
Minor revisions
I would suggest shortening the introduction to avoid repetition in the discussion section.
The language editing should be revised.
Author Response
Dear reviewer,
we are submitting to you the revised manuscript and the point by point rebuttal for the manuscript. We hope this will fulfill the your requests and allow the publication within your journal.
In their study, Rosato and co-workers analyze the impact of a telehealth approach to cure HCV among Italian people who use drugs.
The research context is very topical, especially given the WHO’s goal of eliminating HCV infection by 2030. Indeed, due to their younger age and the significant risk of viral transmission, people who use drugs act as virus reservoirs and are a problematic category of patients to be tested, linked to care, treated, receive appropriate follow-up, and have equitable access to care.
The study design and statistical analysis are appropriate. The authors adequately describe the methods and present the results that support their conclusion.
Minor revisions
I would suggest shortening the introduction to avoid repetition in the discussion section.
We thank the reviewer for this suggestion and we rewrote the introduction section to avoid repetition in the discussion section.
The language editing should be revised.
In agreement with the reviewer we revised the language editing.

Reviewer 2 Report
In the present study, Rosato and colleagues developed a model of care for the management and treatment of PWUD with chronic HCV infection. As the authors exhaustively described, the implementation of a micro-elimination strategy for global HCV elimination would allow to achieve a more effective screening and treatment of such patients.
The manuscript is well written and meaningful; furthemore, I really appreciated the the social and logistical contextualization of the problem.
Below, some minor issues that need to be addressed before final acceptance.
1) Statistical analysis. Continuous variables are expressed as mean +/- SD and compared by student t-test (parametric) or Mann-Whitney test (non-parametric). Was data normality checked? Which statistical test was used?
2) Table 1. Data such as ALT are have SD approaching or even exceed the mean value (73 +/- 71; 73 +/- 59; 73 +/- 83); considering that biochemical parameters are usually not normally distributed showing relevant skewness, I suggest to report continuous variables as median and IQR and compare them by Mann-Whitney test. Furthemore, provide p values even for not significant results.
3) Table 3. Same as table 1.
Author Response
Dear reviewer,
we are submitting to you the revised manuscript and the point by point rebuttal for the manuscript. We hope this will fulfill the your requests and allow the publication within your journal.
In the present study, Rosato and colleagues developed a model of care for the management and treatment of PWUD with chronic HCV infection. As the authors exhaustively described, the implementation of a micro-elimination strategy for global HCV elimination would allow to achieve a more effective screening and treatment of such patients.
The manuscript is well written and meaningful; furthemore, I really appreciated the the social and logistical contextualization of the problem.
Below, some minor issues that need to be addressed before final acceptance.
1) Statistical analysis. Continuous variables are expressed as mean +/- SD and compared by student t-test (parametric) or Mann-Whitney test (non-parametric). Was data normality checked? Which statistical test was used?
The data normality was checked by Shapiro-Wilk test. We thank the reviewer for this correction. We rewrote the statistical analysis section adding this information.
2) Table 1. Data such as ALT are have SD approaching or even exceed the mean value (73 +/- 71; 73 +/- 59; 73 +/- 83); considering that biochemical parameters are usually not normally distributed showing relevant skewness, I suggest to report continuous variables as median and IQR and compare them by Mann-Whitney test. Furthemore, provide p values even for not significant results.
In agreement with the reviewer we reported the continuous variables as median and IQR and we provided the p values also for the not significant results.
3) Table 3. Same as table 1.
We modified the table 3 as the table 1.
